# Semi-Active Vibration Control of Seat Suspension Equipped with a Variable Equivalent Inertance-Variable Damping Device

Guangrui Luan [1], Pengfei Liu [1], Donghong Ning [1,*], Guijie Liu [1] and Haiping Du [2]

[1] College of Engineering, Ocean University of China, Qingdao 266000, China
[2] School of Electrical, Computer and Telecommunications Engineering, University of Wollongong, Wollongong, NSW 2522, Australia
* Correspondence: ningdonghong@ouc.edu.cn

**Abstract:** The seat suspension has a significant influence on riding comfort in many practical applications, such as heavy duty vehicles, military vehicles, and high-speed crafts. This paper proposes a seat suspension equipped with a variable equivalent inertance-variable damping (VEI–VD) device and a novel semi-active vibration control strategy. The VEI–VD device can control its equivalent inertance and damping by controlling two external resistors in its electric circuit. Especially, the VEI part of the device can store and release vibration energy via the inside flywheel, which enables the seat suspension to have a four-quadrant controllable capability in the available force–velocity diagram, similar to an active system. First, the dynamic model of the VEI–VD device is built, and a prototype is developed and tested to identify the model parameters and verify its characteristics. Then, a semi-active vibration control method is proposed for the VEI–VD seat suspension. The control method uses a sliding mode controller to acquire the desired control force for reducing vibration; then, according to the desired force and system states, the VEI–VD device is tuned by a force-tracking scheme to generate a real force. In the numerical validation, the vibration transmissibility of VEI–VD seat suspension around its natural frequency is tested with different states. The effectiveness of force-tracking control strategies for different types of suspensions is verified. In the random excitation test, the root means square acceleration of the VEI–VD seat is reduced by 30.72% compared with a passive seat. The VEI–VD seat suspension shows great potential in applications.

**Keywords:** semi-active control; seat suspension; variable equivalent inertance; variable damping; four-quadrant controllable capability

## 1. Introduction

Low-frequency seat vibration leads to unsafety and uncomfortableness when the vehicle travels on uneven roads [1]. The vibration caused by road conditions is finally transferred through the chassis and seat to the human body; therefore, research on suppressing seat suspension vibration is essential. Researchers have conducted vibration control experiments using passive, semi-active, and active seat suspensions. A passive suspension applied a spring design with negative stiffness [2], and researchers have carried out multi-criteria optimization for passive seat suspension [3]. As an advanced technology in passive suspension system, air suspension system is designed to improve the vehicles' ride comfort [4]. However, the passive seat suspension cannot provide a controllable damping force; its vibration reduction performance is limited. Besides, active seat suspensions are hard to be widely applied [5] because of their high price, massive energy consumption, and complex system. Hence, the novel structure [6,7] and control strategy [8–10] of semi-active seat suspension has become the focus of the research in the vibration reduction field.

Generally, damping, stiffness, and inertance are considered the basic elements in vibration suppression suspension [11,12]. Thus, the semi-active seat suspension with variable damping(VD) [8,13], variable stiffness(VS) [14], and variable inertance (VI) [15]

capacity has attracted a large number of researchers to its in-depth study. Because the semi-active control system can only output the mechanical energy [16], similar to passive systems, thus it only consumes energy. The output mechanical energy of the active control system can come from electric energy [17], hydraulic energy [18], and other forms; it can output positive work. The study of whether there is an actuator that can achieve the effect of active control in the form of a semi-active device that can output negative power becomes a new direction of research [19]. It can rely on the advantages of the semi-active control system, low cost, and simple control to achieve the performance of an active control system while avoiding the trap of high cost and high energy consumption. According to this idea, the variable equivalent inertance (VEI) device based on electromagnetic variable damping is proposed [20].

The inerter has potential in the field of vibration control. The inertance is the mechanical property of an inerter, which is a two-terminal mechanical device, like the spring and damper [16]. The force generated by the inerter is proportional to the relative acceleration of its two terminals [21]. The passive mechanical networks are investigated and applied in vehicle suspension by employing the inerters, springs and dampers [22,23]. Meanwhile, in [23], a novel semi-active variable inertance device is proposed, which replaces the fixed-inertia flywheel with a controllable-inertia flywheel. The shock absorber with variable damping and inertance characteristics is proposed to improve the performance of the seat suspension [24,25]. And according to the analogy between mechanical and electrical systems [26], electrical characteristics of resistance, inductance, and capacitance correspond to the mechanical components of damping, stiffness, and inertance [27]. An electromagnetic variable inertance and damping seat suspension [28] is proposed, and Ning et al. also proposed a semi-active variable equivalent stiffness and inertance device implemented by an electrical network [29].

Because the vibration energy of the suspension can be stored in the inertial element of the VEI device, it can release the energy by appropriate control strategy. The force generated during the release of inertial energy can output positive power, resulting in the effect of the active actuator. According to this, a novel concept of variable equivalent inertance-variable damping (VEI–VD) seat suspension is proposed in this paper, and a VEI–VD device is designed and tested. The VEI–VD device consists of VEI and VD parts. The VEI part stores the energy and use the energy to generate control force; and the VD part can provide conventional semiactive control when the stored energy is not sufficient. Therefore, the VEI–VD seat suspension can achieve the active system's excellent performance while keeping the semi-active system's advantage of low energy consumption.

The main contributions of this paper are as follows:

- A novel concept of the VEI–VD device is proposed and tested experimentally to verify its controllability.
- A novel semi-active force tracking controller is designed to achieve better seat vibration control.

The rest of the paper is organized as follows: Section 2 proposes a novel concept of VEI–VD device and establishes its dynamics model; Section 3 presents a VEI–VD seat suspension prototype; and the force tracking performance of the novel device is discussed in Section 4; Section 5 presents the validation on vibration isolation performance of the VEI–VD seat suspension.; Finally, Section 6 presents the conclusions of this research.

## 2. VEI–VD Seat Suspension System

### 2.1. Motivation

Active seat suspension has the disadvantage of being bulky and expensive, and the passive system has limited performance. Thus, semi-active seat suspension has been widely studied in vibration reduction. Researchers have developed a kind of VEI device consisting of a VD device and two inertial components. By varying the damping of the VD device, the equivalent inertance of the VEI device is controllable in real-time. The VEI device can store energy in the flywheel and release it to suppress vibration. An in-depth study of

the VEI device shows that this part of the energy released from the flywheel can output positive power, similar to the energy characteristic of the active control system. However, the amplitude of energy stored in the flywheel is unstable, which may lead to significant vibration at low frequencies. Therefore, a VD device is applied to assist the VEI device, forming a variable equivalent inertance-variable damping (VEI–VD) suspension.

The VEI–VD device, combined with the conventional passive suspension, forms a VEI–VD seat suspension, as shown in Figure 1, where M is the mass of the body and seat; $K$ and $C$ are the spring stiffness and equivalent damping generated by friction, respectively; $Z_s$ and $Z_v$ are the displacements of the seat and vehicle cab floor, respectively.

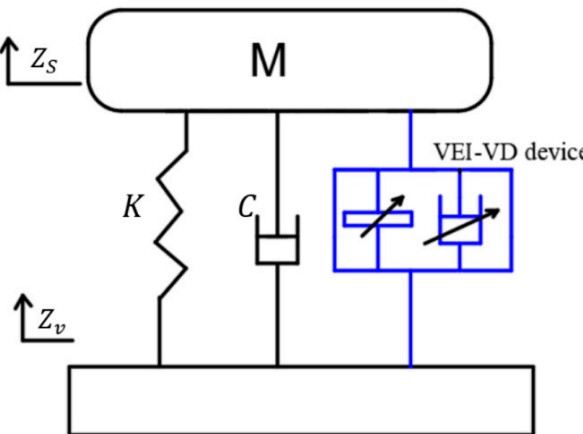

**Figure 1.** The model of VEI–VD seat suspension.

### 2.2. System Analysis

The concept of VEI–VD seat suspension is a novel system that uses the positive power generated by the VEI device and cooperates with the VD device to produce an effect similar to the active control system. In addition, it takes into account the advantages of semi-active control seat suspension, such as low energy consumption, simple structure, and high reliability.

The VEI device consists of a ball screw, a rotary VD device, and a flywheel, as shown in Figure 2. $C_1$ is the damping of the VD device. $J$ is the moment of inertia of flywheel. $Z_v$ and $Z_s$ are the displacement of two terminals, and $\alpha$, $\beta$ are the rotation angle of ball screw and flywheel, respectively. The function of the ball screw is changing the reciprocating motion of vibration into rotary motion, and by varying the damping of the VD device, the equivalent inertance of the VEI device is controllable in real-time. In the ideal status, when the damping of the VD device is zero, the flywheel is in a state of free rotation and is disconnected from the ball screw. When the damping of the VD device is infinite, the flywheel considers that it is in a state of fixed connection with the ball screw. Therefore, it can be considered that the energy output of the whole device will change when $C_1$, the damping of the VD device in the VEI device, is adjusted. Generally, the harvested vibration energy by the ball screw can be stored in the flywheel of the VEI device. And, the energy stored in the flywheel can be released under certain circumstances. The process of releasing energy can be tuned through appropriate control strategies to cope with vibrations.

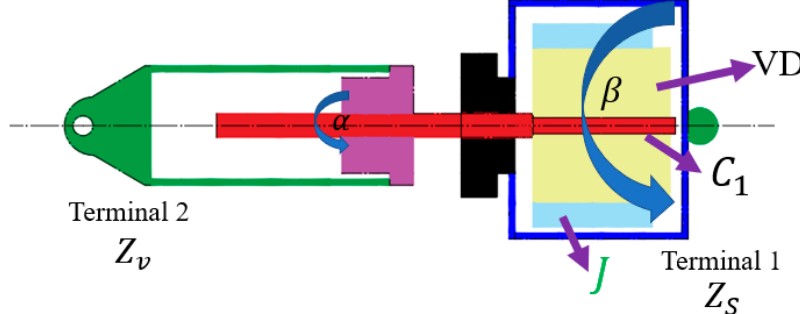

**Figure 2.** The model of the VEI device.

The dynamic model of the VEI device can be built with the consideration of the flywheel [15]:

$$C_1\left(\dot{\alpha} - \dot{\beta}\right) = J\ddot{\beta}, \tag{1}$$

$$\alpha = R(Z_s - Z_v), \tag{2}$$

where $R$ is the transformation ratio of the ball screw, and $R = 2\pi/d$, the $d$ is the lead of the ball screw.

The force generated by the VI device:

$$F_b = R^2 C_1(\dot{\alpha} - \dot{\beta}), \tag{3}$$

The VD device uses a ball screw to convert the reciprocating motion of vibration into rotary motion, and the rotation angle of the ball screw is $\alpha$ also. The $C_2$ of the VD device can be adjusted to change the output of the VD device. In the VD device, the force is:

$$F_D = R^2 C_2 \dot{\alpha}, \tag{4}$$

The VEI–VD device is composed of the VEI device and VD device. Thus, the output force of the VEI–VD device:

$$F_{out} = F_b + F_D, \tag{5}$$

where $F_b$, $F_D$, and $F_{out}$ are the force generated by the VEI, VD and VEI–VD device in theory, respectively. This model analyses the device's controllability and simplifies the nonlinear factors in the system, such as the inherent inertia of the ball screw and the friction force of the system. The subsequent experiment tests will determine these nonlinear factors to supplement the model.

According to the above dynamic equation, we can get the device admittance:

$$Y_i = R^2(C_1 + C_2) - \frac{R^2 C_1{}^2}{J^2 \omega^2 + C_1{}^2} + \frac{R^2 J C_1{}^2}{J^2 \omega^2 + C_1{}^2} j\omega, \tag{6}$$

where $R^2(C_1 + C_2) - \frac{R^2 C_1{}^2}{J^2\omega^2 + C_1{}^2}$ represents the equivalent linear damping, and $\frac{R^2 J C_1{}^2}{J^2\omega^2 + C_1{}^2}$ is the equivalent linear inertance of device.

In order to analyze the frequency characteristics of the VEI–VD device, we apply a set of parameters to $Y_i$, where $R = 2\pi/0.016$, $J = 160 \times 10^{-5}$ kgm$^2$, $C_{min}$ and $C_{max}$ are the minimum and maximum damping of the variable dampers, respectively. $C_1$ and $C_2$ are assumed to be controllable within $C_{min} = 2.64 \times 10^{-3}$ Nms/rad to $C_{max} = 1.5 \times 10^{-2}$ Nms/rad, the VEI and VD devices use electromagnetic variable dampers of the same design, and the damping value can be changed from minimum to maximum.

The Figure 3 shows that when the $C_2$ is set with its minimum value and $C_1$ varies in its controllable range, with the increasing of $C_1$, the equivalent linear inertance of VEI–VD is increasing. When the damping $C_1$ is constant, the equivalent inertance decreases with the increase of $\omega$. The frequency in the X-axis is $2\pi/\omega$. The reason is that, in the

traditional inerter, the rotation of flywheel in the inerter will generate the inertance force, while in the proposed device, the flywheel is driven by the controllable damping torque, and the flywheel needs time to accelerate to a high velocity. When the frequency of the reciprocating rotation is high, the flywheel has not time to accelerate to a high velocity, thus, the equivalent inertance of the device is low. In addition, the equivalent linear damping of the VEI–VD device is determined by VEI and VD devices. Obviously, with the increase of the $C_2$, the equivalent linear damping is increasing and the Figure 4 shows that the equivalent damping of the device is controllable with $C_1$. Thus, the equivalent linear damping will also increase with the increase of the two dampers. The results of the simulation indicate that the inertance and damping of the VEI–VD device can be controlled by varying the magnitude of $C_1$ and $C_2$.

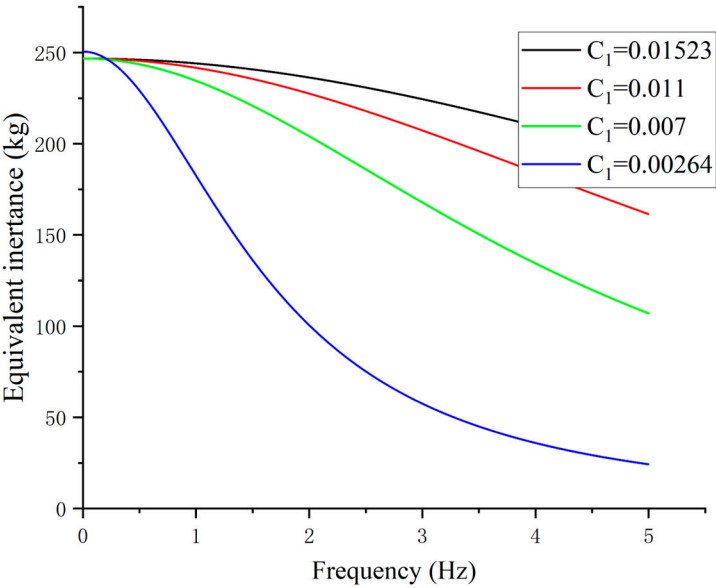

**Figure 3.** The equivalent inertance of VEI–VD.

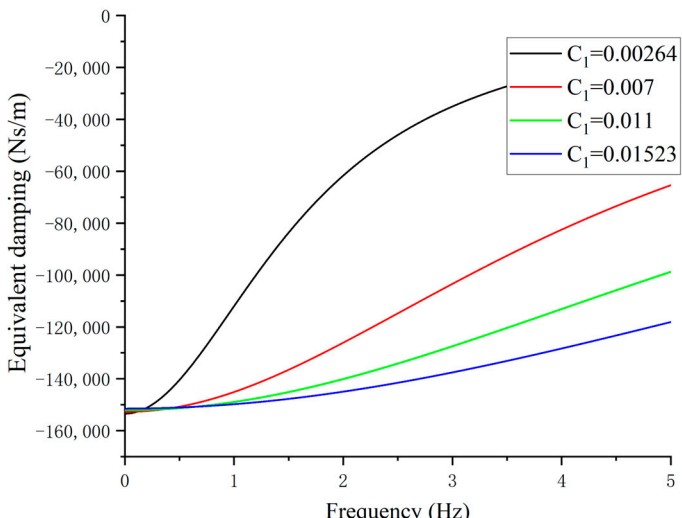

**Figure 4.** The equivalent damping of VEI–VD.

## 3. VEI and VD Device Test

### 3.1. Prototype of VEI and VD Device

An electromagnetic VD device using a permanent magnet synchronous motor (PMSM) has been proposed in [30]. Due to the excellent controllability of the electromagnetic damping device (EMD) demonstrated in the experiments, the VEI–VD device adopts two

identical PMSM (MSMD022G1S), and the inertial element of the VEI device is a flywheel. In the VEI device, the ball screw is connected to the shaft of the motor through a coupling, and the flywheel is attached to the shell of the PMSM. The flywheel rotates with the shell of the PMSM. The wire of the PMSM is led out through the conductive slip ring for its control. The prototype design is shown in Figure 5.

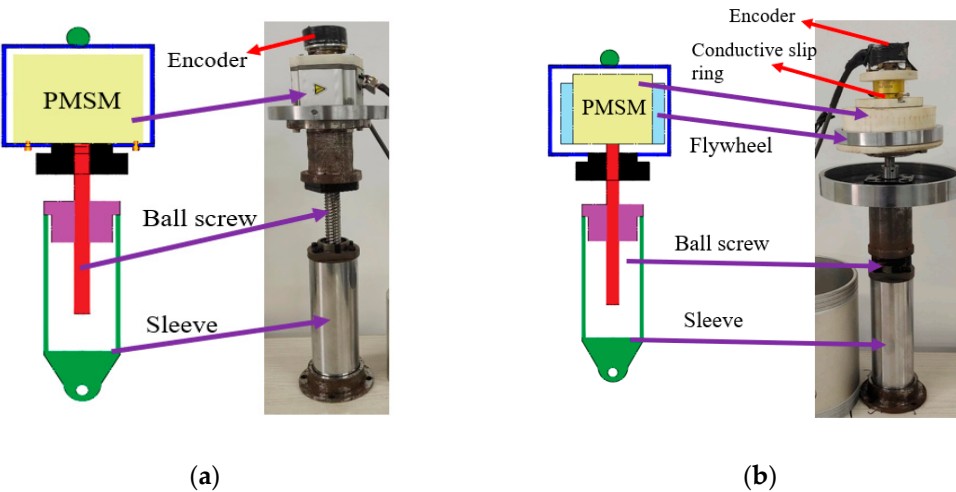

(**a**)　　　　　　　　　　　　　　　　(**b**)

**Figure 5.** Prototype of VEI–VD device: (**a**). VD device; (**b**). VEI device.

A real-time variable resistance circuit is designed to realize the function of the VD device [29], which is shown in the Figure 6. A three-phase rectifier is used to convert the three-phase electricity of PMSM into direct current. The PMSM and rectifier are equivalent to a voltage source $e_i$, an internal resistor $R_i$, and an internal inductor $L_i$. The internal inductance $L_i$ is ignored to simplify the modelling [30] because the seat vibration energy is mainly at low frequency where the inductor can be treated as a conducting wire. At the same time, a controller (NI My-Rio 1900) is used to control the external resistance. Thus, the magnitude of resistors connected to the circuit can be controlled, achieving the purpose of controlling the resistance of the whole circuit.

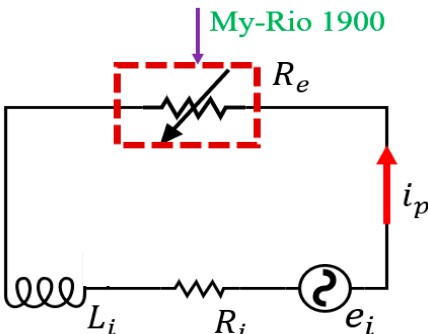

**Figure 6.** Simplified circuit model of EMD.

Theoretically, the damping of the VD device can be changed by controlling the total resistance of the PWSM circuit [30]. The generated voltage is proportional to the rotary rate of the PMSM $\omega$ with a voltage constant $K_e$,

$$e_i = K_e \omega, \tag{7}$$

The current $i_p$ will make the PMSM generate a torque,

$$\text{T} = K_i i_p = K_i \frac{e_i}{R_e + R_i} = \frac{K_i K_e}{R_e + R_i} \omega, \tag{8}$$

where $K_i$ and $K_e$ are the voltage and current constant of PMSM, respectively. It is generally believed that $K_i = K_e$.

Then, the controllable rotary damping of the VD device is:

$$C_T = \frac{K_i K_e}{R_e + R_i},$$ (9)

where $C_T$ is the real-time damping of the VD device; $R_i$, $R_e$ are the internal resistance of PMSM, external resistance, respectively.

### 3.2. Test Design

The time-domain test can effectively characterize the damping and inertance properties of the VEI–VD device, and verify the accuracy of the model. According to the time-domain test, it can be used to develop the model of VEI–VD. The time-domain test bench is shown in Figure 7, where the servo motor driven (MSMF082L1A1) electric cylinder provides the power source at the bottom of the bench, a laser displacement sensor (KathMatic 200 mm) is applied to measure the displacement of device, and a force sensor (Transcell FAK-250 kg) was installed on the upper part of the test bench. In addition, NI My-Rio 1900 is the control center of the whole test system, and its tasks include controlling the servo motor rotating following a set curve, controlling the resistance change of the EMD circuit, receiving voltage signals of the force sensor and displacement sensor and converting them into corresponding physical quantities.

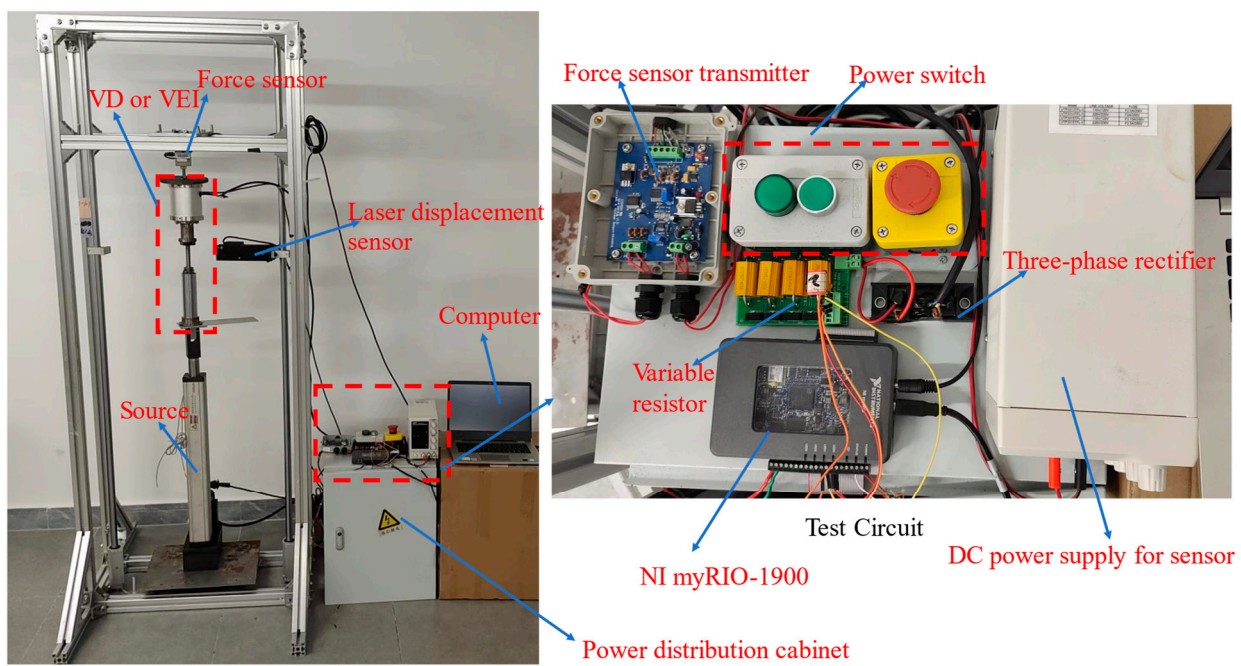

**Figure 7.** Time-domain test bench.

### 3.3. Time-Domain Characteristic

The VEI and VD devices were verified separately. A sine excitation source x = A ∗ $\sin(2\pi f t)$ is applied to the device, where $A = 0.01$ m, $f = 1.5$ Hz. In the test, the branch resistors $R_e$ are 0, 3, 8, 15, 50 Ohm, respectively.

In the experiments, it was found that the main uncontrollable factors affecting the VEI device included the friction between the device parts and the inherent inertial of the shaft and PMSM itself.

$$F_{j1} = R^2 J_1 (\ddot{Z}_s - \ddot{Z}_v),$$ (10)

$$F_{r1} = f_{r1} sat \left( \dot{Z}_s - \dot{Z}_v \right),$$ (11)

where the $sat\left(\dot{Z}_s - \dot{Z}_v\right)$ is defined:

$$sat\left(\dot{Z}_s - \dot{Z}_v\right) = \begin{cases} 1 & \left(\dot{Z}_s - \dot{Z}_v\right) > \tau \\ \frac{1}{\tau}(\dot{Z}_s - \dot{Z}_v) & -\tau \le \left(\dot{Z}_s - \dot{Z}_v\right) \le \tau, \\ -1 & \left(\dot{Z}_s - \dot{Z}_v\right) < -\tau \end{cases} \tag{12}$$

where $\tau$ can be determined according to the $f_{r1}$.

Thus, the output force of the VEI device can be obtained:

$$F_b = R^2 C_1 (\dot{\alpha} - \dot{\beta}) + F_{j1} + F_{r1}, \tag{13}$$

Figure 8 shows the characteristics of the VEI device. The VEI output force and the displacement exhibit negative stiffness characteristics, and when the total resistance increases, the negative stiffness decreases, which means that the inertance of the VEI device decreases. The result indicates the method of changing the external resistance can adjust the inertance of the VEI device.

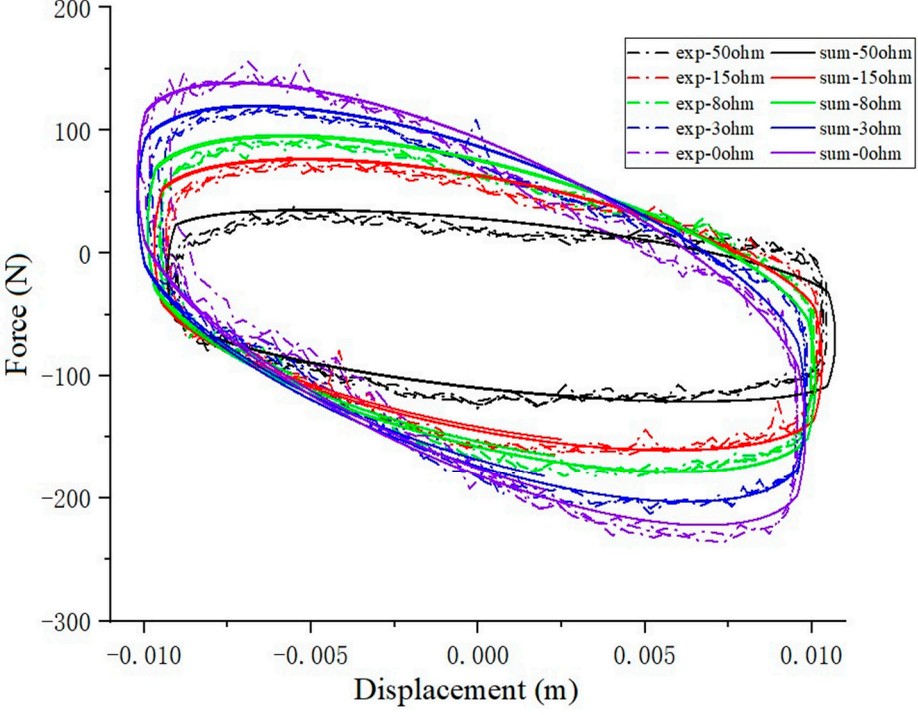

**Figure 8.** Force-displacement characteristics of VEI device.

For the VD device, the friction between the device parts and the inherent inertial of the ball screw itself affect the system.

$$F_{j2} = R^2 J_2 (\ddot{Z}_s - \ddot{Z}_v), \tag{14}$$

$$F_{r2} = f_{r2} sat\left(\dot{Z}_s - \dot{Z}_v\right), \tag{15}$$

The output force of the VD device is:

$$F_D = R^2 C_2 \dot{\alpha} + F_{j2} + F_{r2}, \tag{16}$$

Figure 9 shows that the variable damping characteristic of the VD device under different external resistance. It is seen that the enclosed areas of the force–displacement

loops increase with the increasing of external resistance, which means that the damping coefficient can be controlled by changing the branch resistor. The results of the VEI–VD characteristic curve show that the device can use the controller to select appropriate branch resistors to adjust the damping and inertance.

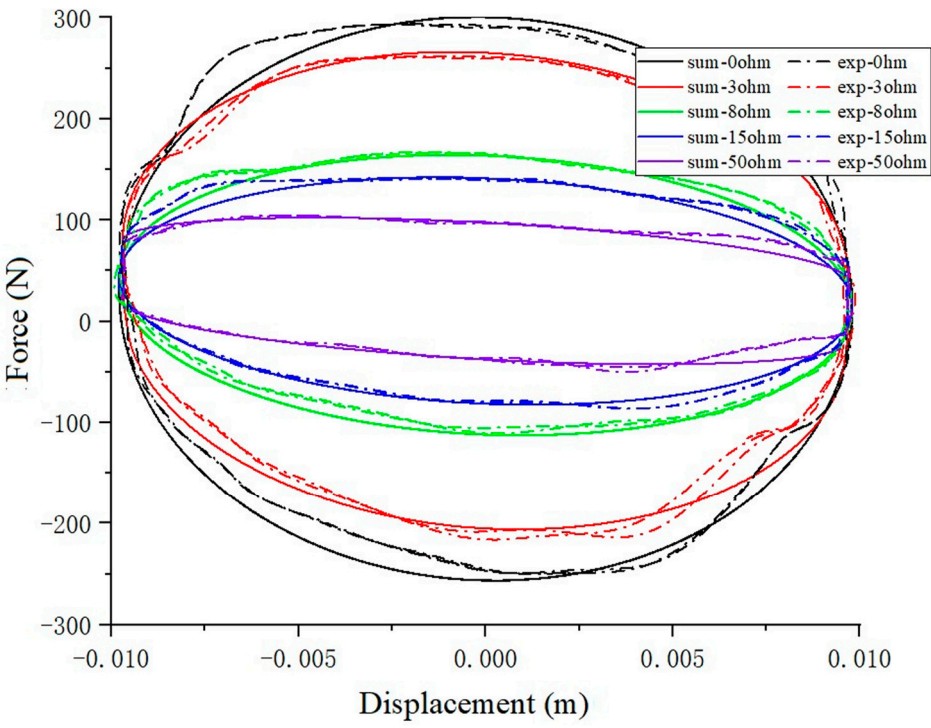

**Figure 9.** Force-displacement characteristics of VD device.

A series of parameters of VEI–VD device are determined through the experiment, as shown in Table 1. After the optimization of the system, the uncontrollable output force of the VEI–VD device is:

$$F_r = f_{r1} sat\left(\dot{Z}_s - \dot{Z}_v\right) + f_{r2} sat\left(\dot{Z}_s - \dot{Z}_v\right), \tag{17}$$

$$F_j = R^2 J_1(\ddot{Z}_s - \ddot{Z}_v) + R^2 J_2(\ddot{Z}_s - \ddot{Z}_v), \tag{18}$$

where $f_{r1}$, $f_{r2}$, are the friction coefficient of VEI and VD device respectively, and $F_r$ is friction force of the VEI–VD device; $J_1$, $J_2$, are the inherent moment of inertia of VEI and VD device respectively, and $F_j$ are moment of inertia force of the two devices.

**Table 1.** Parameters of the VEI–VD device.

| Semiactive Device | Parameter | Symbol | Value |
|---|---|---|---|
| | Friction coefficient | $f_{r1}$ | 35 N |
| VEI device | Inherent moment of inertia | $J_1$ | $200 \times 10^{-7}$ kgm$^2$ |
| | Moment of inertia of flywheel | $J$ | $160 \times 10^{-5}$ kgm$^2$ |
| | Friction coefficient | $f_{r2}$ | 30 N |
| | Inherent moment of inertia | $J_2$ | $150 \times 10^{-7}$ kgm$^2$ |
| VD device | Constant of PMSM | $K_i(K_e)$ | 0.41 Nm/A(Vs/rad) |
| | Internal resistance of PMSM | $R_r$ | 10.5 Ohm |

The output force of the VEI–VD device is expressed as:

$$F_{out} = F_D + F_b, \tag{19}$$

## 4. Controller Design

### 4.1. Control Scheme

For the VEI–VD seat suspension system, the semi-active force tracking control of the control scheme is adopted. The flow chart of the control scheme is shown in Figure 10. The $R_1$ and $R_2$ are the branch resistors of the VEI and VD device, respectively.

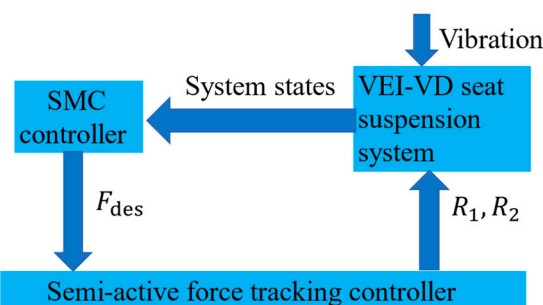

**Figure 10.** Control scheme.

### 4.2. Sliding Mode Control

The active control strategy adopts the sliding mode control. The VEI–VD seat suspension model can be built as:

$$M\ddot{Z}_s = -K(Z_s - Z_v) - F_{out},\tag{20}$$

where $M$ is the mass of the body and seat; $K$ is the stiffness of the spring; $F_{out}$ is the generated VEI–VD device force.

We assume an active seat suspension for designing the controller:

$$M\ddot{Z}_s = -K(Z_s - Z_v) - f_{des} - F_j - F_r,\tag{21}$$

where $f_{des}$ is the desired control force, which will be provided by the VEI–VD seat suspension.

The tracking error of the controller is [30]:

$$e = Z_{des} - Z_s,\tag{22}$$

where $Z_{des}$ is the desired displacement, which is assumed as zero.

Therefore, the sliding surface is defined as:

$$s = ce + \dot{e},\tag{23}$$

where $c > 0$.

The derivative of the sliding surface is:

$$\dot{s} = \frac{K(Z_s - Z_v) + f_{des} + F_j + F_r}{M} + c\dot{e},\tag{24}$$

Considering the Lyapunov functions as:

$$L = \frac{1}{2}s^2,\tag{25}$$

So, its derivative is:

$$\dot{L} = s\dot{s} = s * \left[ \frac{K(Z_s - Z_v) + f_{des} + F_j + F_r}{M} + c\dot{e} \right],\tag{26}$$

Therefore, the sliding mode controller can be designed as:

$$f_{des} = -K(Z_s - Z_v) - F_j - F_r - \gamma sgn(s) - Mc\dot{e}, \tag{27}$$

where $\gamma > D$, the $D$ *is* disturbance upper bound.

Last, we can get:

$$\dot{L} = \frac{-s\gamma sgn(s)}{M} = -\frac{\gamma \mid s \mid}{M} \leq 0, \tag{28}$$

When the $\dot{L} \equiv 0$, $s \equiv 0$, The system becomes stable. And as time goes on, $s$ converges and the system is stable.

In the traditional SMC, the chattering phenomenon will happen because the sign function in the controller may lead to a significant change of control force. Thus, to avoid the chattering phenomenon [30], $sgn(s)$ is replaced by $sat(s)$ in the implementation of the controller. When the sign function is replaced by the saturation function, the convergence speed of the controller may be influenced because the force in the saturation term needs time to reach the settled saturation value. And, we have done further discussion of the saturation function.

$$sat(s) = \begin{cases} 1 & k > \frac{1}{\Delta} \\ ks & k = \frac{1}{\Delta} \\ -1 & k > \frac{1}{\Delta} \end{cases}, \tag{29}$$

where $\Delta$ is the "boundary layer". The application of the boundary layer can be further deepened. Gohari et al. propose a novel strategy nominated as a self-adjusting boundary layer in order to prevent the occurrence of the chattering phenomenon [10].

And, we can get a result:

$$\dot{L} = \frac{-s\gamma sat(s)}{M} = \begin{cases} -\frac{\gamma|s|}{M} \leq 0 & k > \frac{1}{\Delta}, \\ -\frac{k\gamma s^2}{M} \leq 0 & k = \frac{1}{\Delta} \end{cases}, \tag{30}$$

*4.3. Force Track Control Strategy*

The traditional semi-active control strategy of VEI and VD device may have superiority in some states; the semi-active force tracking control scheme of VEI and VD device is shown in Figure 11.

Firstly, the VD device is assigned to track desired force:

$$C_{VD} = \frac{f_{VDdes}}{\left(R^2 \alpha_{VD}^{\cdot}\right)}, \tag{31}$$

where $C_{VD}$ is the assigned damping of the VD device, the $f_{VDdes}$ is the desired force of its ideal controllers, and the $\alpha_{VD}^{\cdot}$ is the shaft rotary speed of the VD device.

Similarly, the traditional VEI device is assigned to track desired force:

$$C_{VEI} = \frac{f_{VEIdes}}{\left(R^2 \left(\alpha_{VEI}^{\cdot} - \dot{\beta}\right)\right)}, \tag{32}$$

where $C_{VEI}$ is the assigned damping of the VEI device, the $f_{VEIdes}$ is the desired force of its ideal controllers, and the $\alpha_{VEI}^{\cdot}$ is the shaft rotary speed of the VEI device. the ideal controllers are resembling the sliding mode controller in the Section 4.2. The controller scheme is shown in the Figure 11.

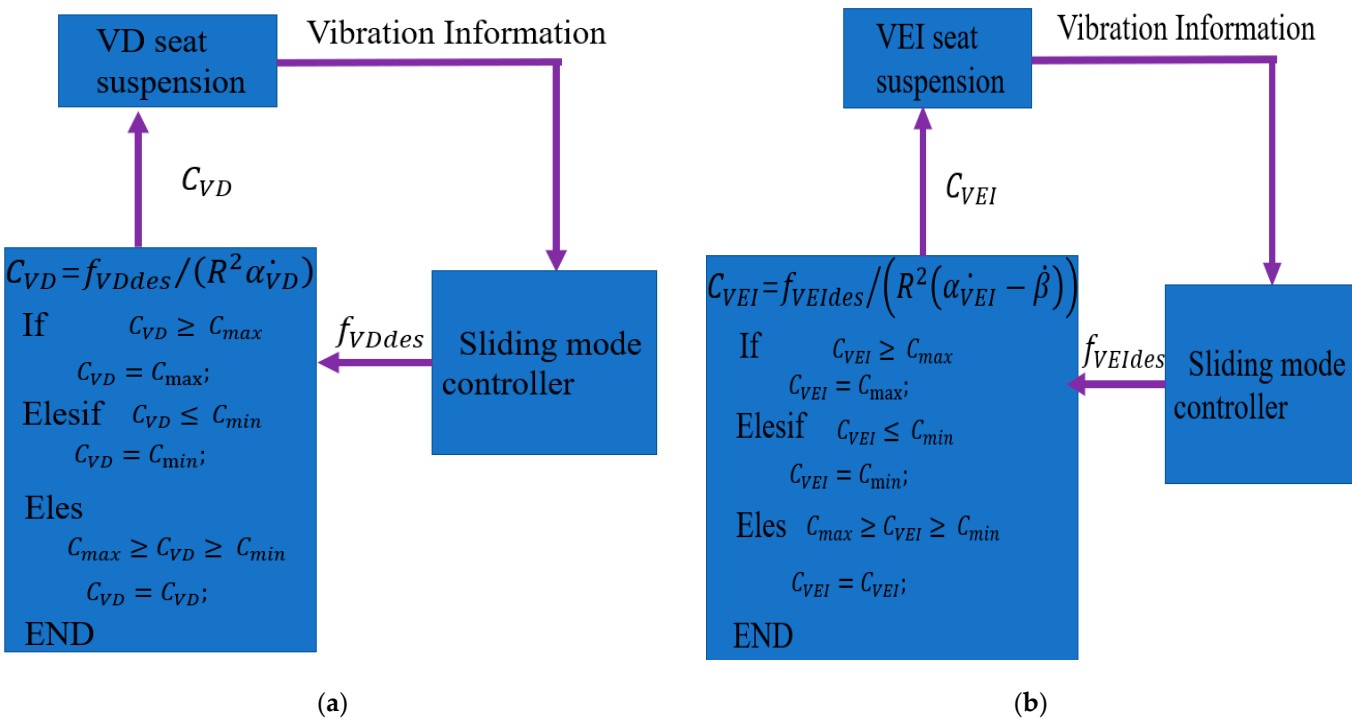

**Figure 11.** The VD and VEI device control scheme: (**a**). The VD device control scheme; (**b**). The VEI device control scheme.

For VEI–VD seat suspension, an appropriate force-tracking control strategy is the most important part. How to use the VEI device to collect vibration energy, release the collected energy, and use the VD device to suppress vibration is the focus of the tracking control strategy. This paper proposes a semi-active control strategy based on the energy storage characteristics of the VEI–VD seat. The flow chart of the execution process of the control strategy is shown in Figure 12. The vibration energy of the seat suspension is stored in the flywheel of the VEI device first. Then, the stored positive power is released at the right moment to suppress the vibration of the seat suspension, which can effectively improve the vibration control performance.

Step 1: The VEI device is working, the energy is stored:

$$C_2 = C_{min} , \tag{33}$$

$$C_1 = \frac{(f_{des} - f_{1min})}{\left(R^2\left(\dot{\alpha} - \dot{\beta}\right)\right)}, \tag{34}$$

$$f_{1min} = C_{min}R\dot{\alpha}, \tag{35}$$

Then, the value of $C_1$ needs to be judged, when $C_1 \leq C_{min}$, the $C_1 = C_{min}$, and when $C_{min} < C_1 < C_{max}$, the $C_1 = C_1$; however, $C_1 > C_{max}$, the VEI device is insufficient, then, the Step 2 starts execution.

Step 2: $C_1$ of the VEI device is maximum, the VD begins to work; the energy is released:

$$C_1 = C_{max}, \tag{36}$$

$$C_2 = \frac{(f_{des} - f_{2max})}{\left(R^2\dot{\alpha}\right)}, \tag{37}$$

$$f_{2max} = C_{max}R^2\left(\dot{\alpha} - \dot{\beta}\right), \tag{38}$$

when $C_2 \leq C_{min}$, the $C_2 = C_{min}$, when $C_{min} < C_2 < C_{max}$, the $C_2 = C_2$, and when $C_1 > C_{max}$, the $C_2 = C_{max}$.

Step 3: The basic principle of the electromagnetic damper is used, and the appropriate external resistance value is calculated.

$$R_1 = \frac{K_i * K_e}{C_1} - R_i, \tag{39}$$

$$R_2 = \frac{K_i * K_e}{C_2} - R_i, \tag{40}$$

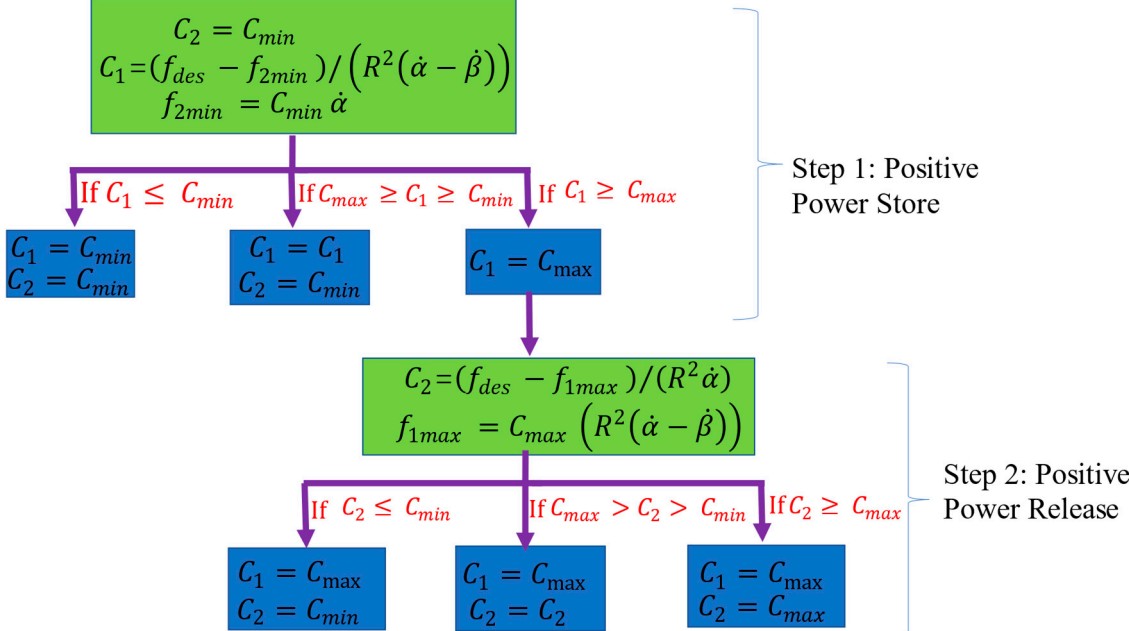

**Figure 12.** NEPS control strategy.

## 5. Numerical Validation

### 5.1. Feasibility Test

To further verify the feasibility of VEI–VD seat suspension, the relevant tests were carried out. The first thing to do is to determine the full working range of the VEI–VD seat suspension. In the test, the suspension is divided into four states, including (1).the external resistance of the VEI–VD device is connected to 50 Ohm, (2).the VD device is connected to 0 Ohm, and the VEI is connected to 50 Ohm, (3).the VEI device is connected to 0 Ohm and the VD is connected to 50 Ohm, (4).the VEI–VD device is connected to 0 Ohm. A sine source $x = A * \sin(2\pi f t)$ is applied, $A = 0.02$ m, $f = 1.5$ Hz, the acceleration vibration transmissibility of the four states near the resonance point of the system is tested. The results, Figure 13, tests show that with the change of the system state, the acceleration transmissibility of the suspension is changing, and it implies that the VEI–VD suspension adopting the method of changing the external resistance is feasible in terms of controllability.

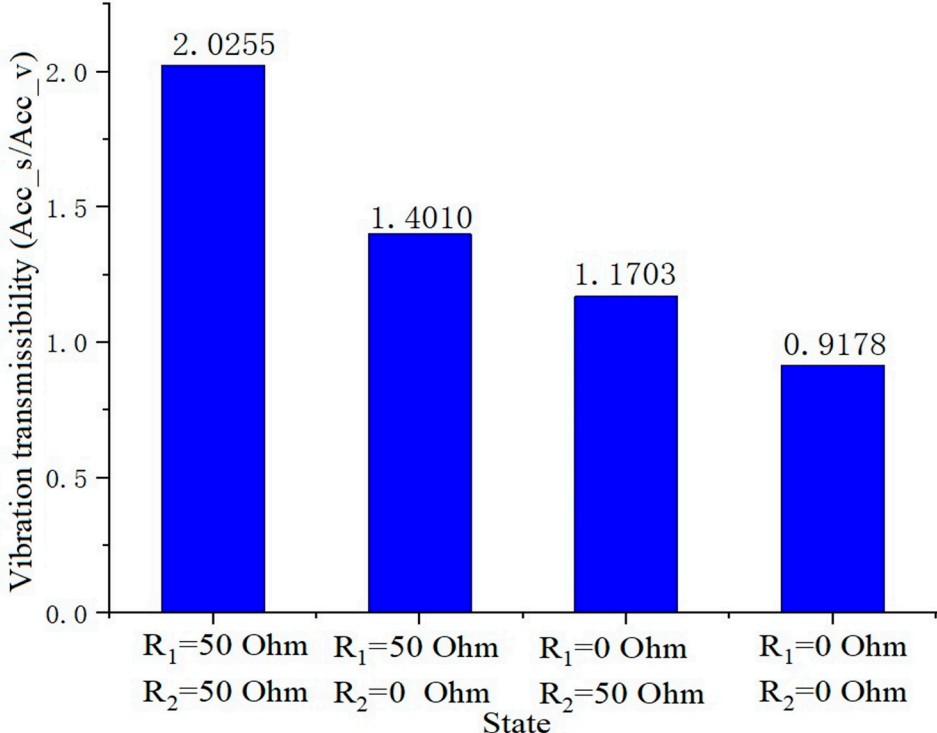

**Figure 13.** Acceleration transmissibility of suspension.

The sinusoidal source is used to test sliding mode control and semi-active force tracking control strategies. The performance of force tracking control strategies is shown in Figure 14, which shows the desired force and the force of the VD, VEI, and VEI–VD suspension systems. The force of the VEI–VD suspension seems to have not tracked the desired one well, because the stored energy of VEI equipment is unstable. If there is an additional active power to supply the desired power totally, the system can have its best performance. How to continue to increase this energy is the focus of the next step. The proposed system can control the VEI–VD device to track a part of the desired force, and hence, generate a beneficial force to improve the seat suspension's performance in vibration isolation.

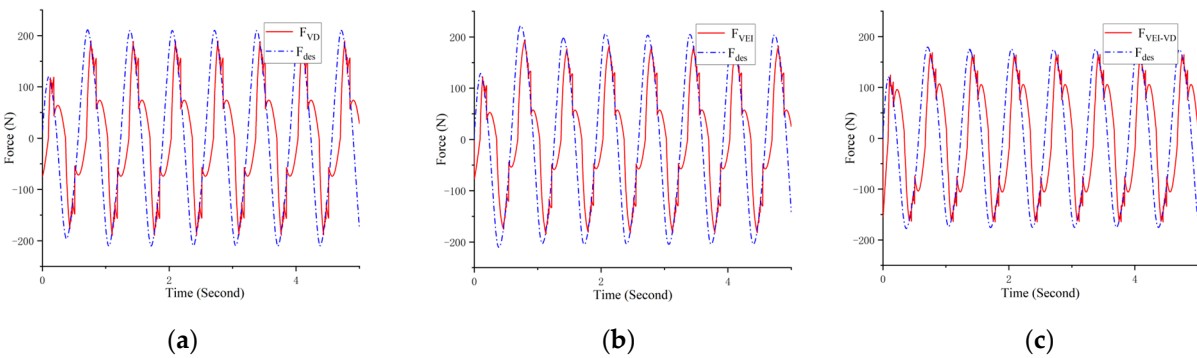

| (a) | (b) | (c) |

**Figure 14.** The performance of force tracking strategies: (**a**) The force tracking of the VD device; (**b**) The force tracking of the VEI device; (**c**) The force tracking of the VEI–VD device.

The simulation experiment of VEI and VD device was carried out, Figure 15 shows that the VD device can only achieve a limited region in the second and fourth quadrants of the available force-velocity diagram, while the VEI device can achieve mechanical control within the four quadrants of the available force-velocity diagram. However, the VEI–VD device loses a part of positive power compared with VEI device because the VD device

plays a part. More essentially, from the energy point of view, both VEI device and VEI–VD device can store energy.

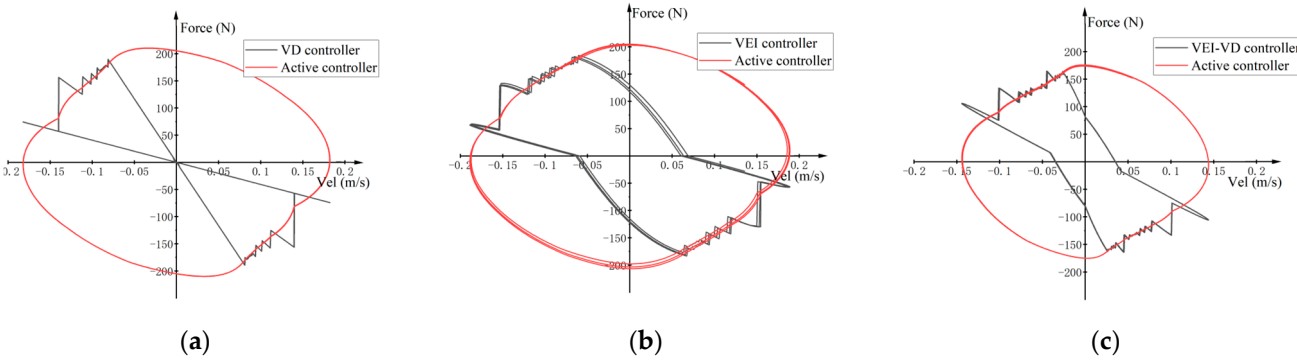

**Figure 15.** Available force-velocity areas of actuators: (**a**) VD device; (**b**) VEI device; (**c**) VEI–VD device.

### 5.2. Sinusoidal Excitation Test

The test employs four different frequencies sinusoidal excitation to verify the performance of the three control methods in Figure 16. The excitations are defined as $x = A * \sin(2\pi f t)$, where $A = 0.02$ m and $f_1 = 1.0$ Hz, $f_2 = 1.5$ Hz, $f_3 = 2.0$ Hz, $f_4 = 2.5$ Hz. The $f_2$ approximates the natural frequency of the seat suspension. The test can estimate the optimal vibration isolation range of the three seat suspensions. Figure 16 shows that VD, VEI, and VEI–VD seat suspension have obvious advantages compared with passive seat suspension. The Table 2 shows the acceleration root mean square (RMS) under different circumstances, which directly indicates that VD, VEI, and VEI–VD seat suspension have their own advantages under different frequencies. When the frequency is lower than 1.5 Hz, the VEI–VD has the best vibration isolation performance, and with the increasing of the excitation frequency, the VEI device gradually shows advantages compared with VEI–VD. However, the VD devices show superior performance when the excitation frequency is higher than the 2.5 Hz. The sinusoidal excitation test shows that the three devices and their control methods have different advantages under different excitation frequencies. In the vehicle seat excitation, the vibration below 2.0 Hz accounts for the majority, thus the comprehensive performance of VEI–VD equipment is superior.

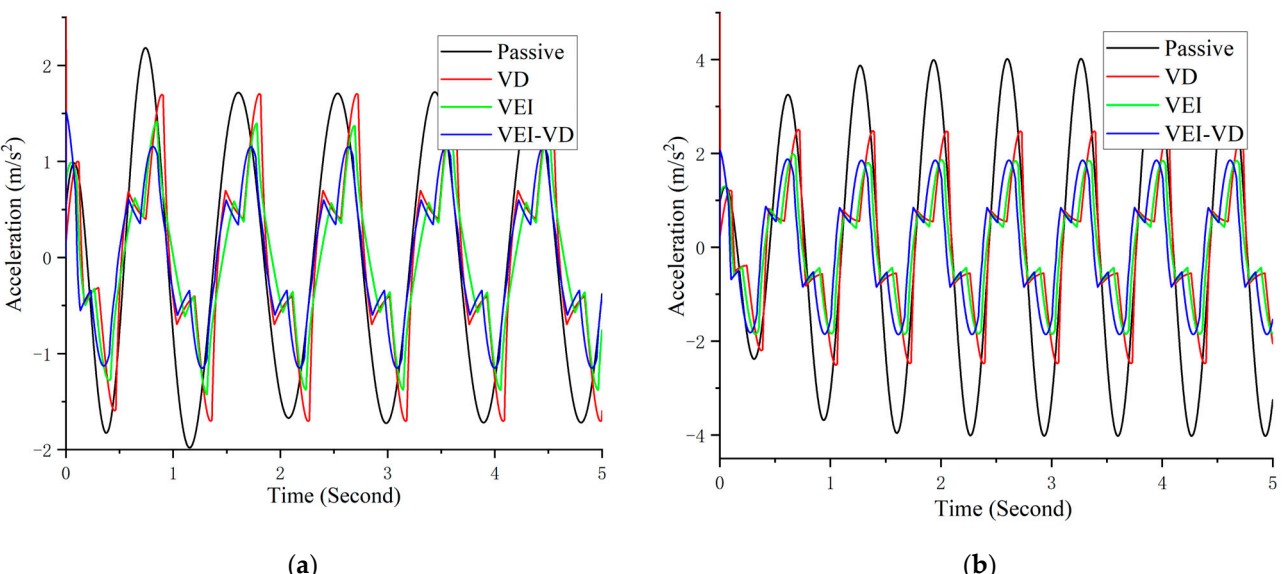

**Figure 16.** *Cont.*

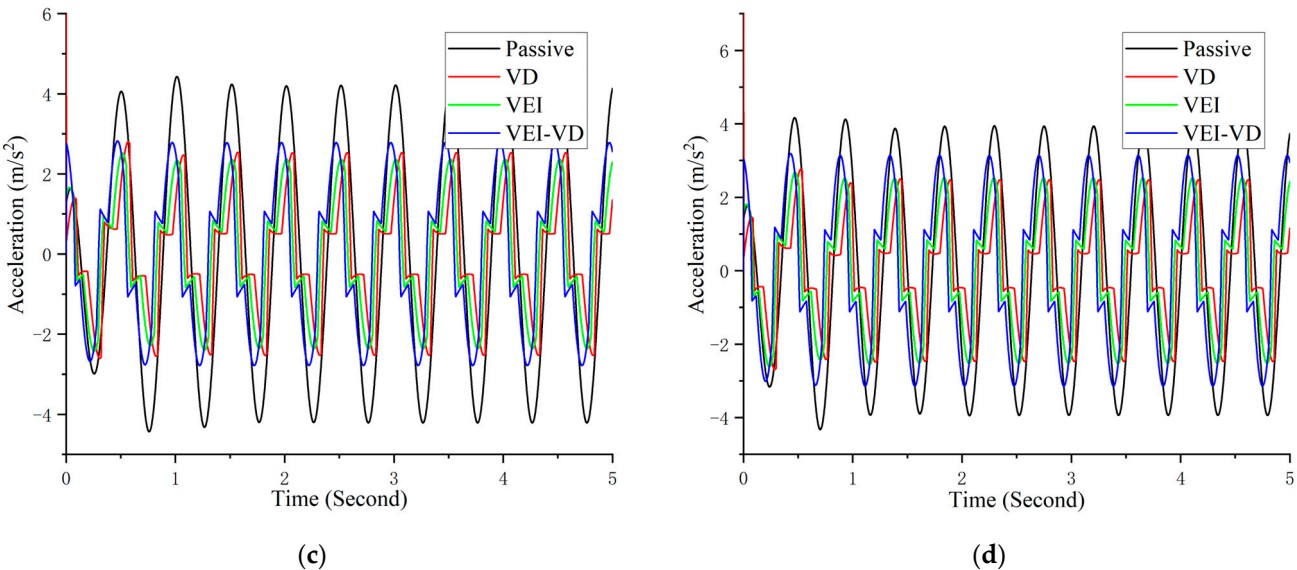

**Figure 16.** Acceleration under different sinusoidal excitation: (**a**) $f_1 = 1.0$ Hz; (**b**). $f_2 = 1.5$ Hz; (**c**). $f_3 = 2.0$ Hz; (**d**). $f_4 = 2.5$ Hz.

**Table 2.** The RMS acceleration reduction of different sinusoidal excitation.

| Device | 1.0 Hz | 1.5 Hz | 2.0 Hz | 2.5 Hz |
|---|---|---|---|---|
| Passive | — | — | — | — |
| VD | 10.72% | 50.67% | 51.69% | 49.37% |
| VEI | 27.64% | 60.13% | 48.89% | 40.97% |
| VEI–VD | 33.07% | 59.13% | 35.87% | 22.56% |

*5.3. Radom Vibration Test*

For evaluating its time-domain performance, a typical road condition is used to test seat suspension performance. The displacement of the sprung mass of the quarter-car model is taken as the vibration input to the seat suspension.

The seat acceleration comparison of the four kinds of seat suspensions is shown in Figure 17; it shows the high magnitude peak of the acceleration is successfully reduced. The frequency analysis in Figure 18 indicates that all seat suspensions can isolate vibration in frequencies higher than 4.0 Hz. When the frequency is lower than 3 Hz, the vibration isolation performance of the VEI–VD seat suspension with semi-active control is obviously better than that of the other.

Based on ISO 2631-1, the frequency weighted-root mean square (FW-RMS) acceleration and the fourth power vibration dose value (VDV) are obtained to evaluate the seat suspensions' performance. In Figure 19, the reduction of the FW-RMS indicates the improvement of ride comfort, and the VDV values show that the VEI–VD seat suspension has a superior performance in the shock vibration. Table 3 shows the performance improvement of the three semi-active suspensions compared with the passive suspension. From the effect of the random excitation test, the performance effect of VEI–VD seat suspension is better than that of VD and VEI seat suspension, which verifies the views in 5.2 Sinusoidal excitation test.

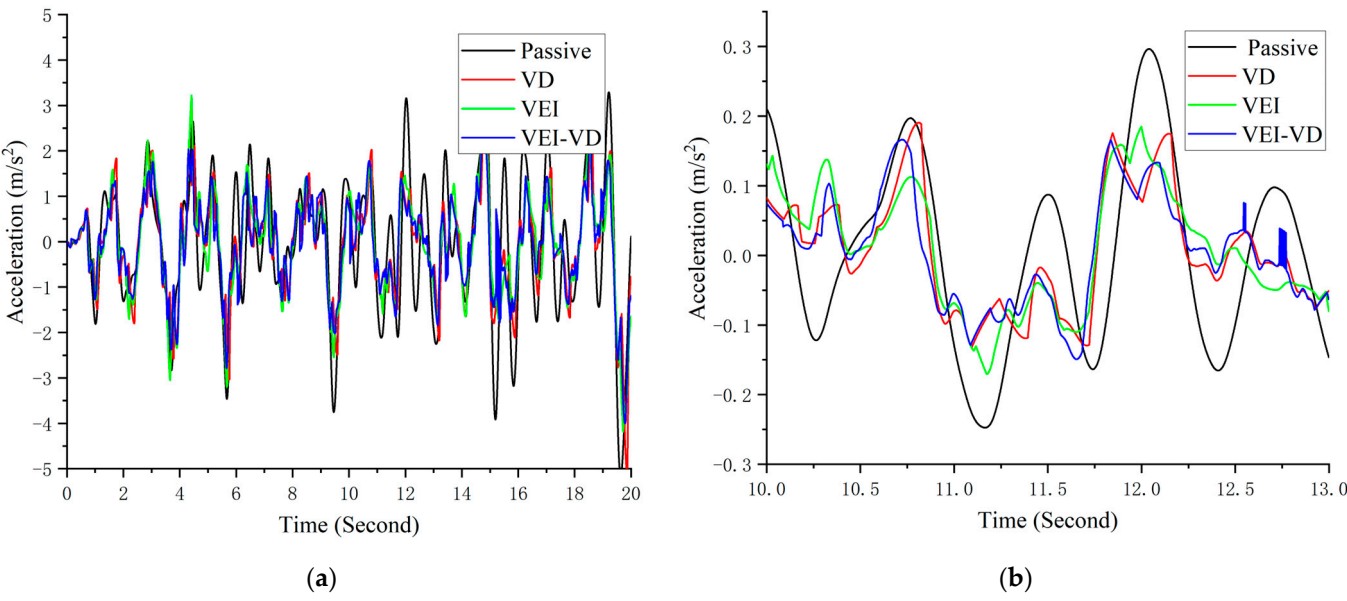

**Figure 17.** Seat acceleration in the time domain: (**a**). Entire experiment; (**b**) Zoom in.

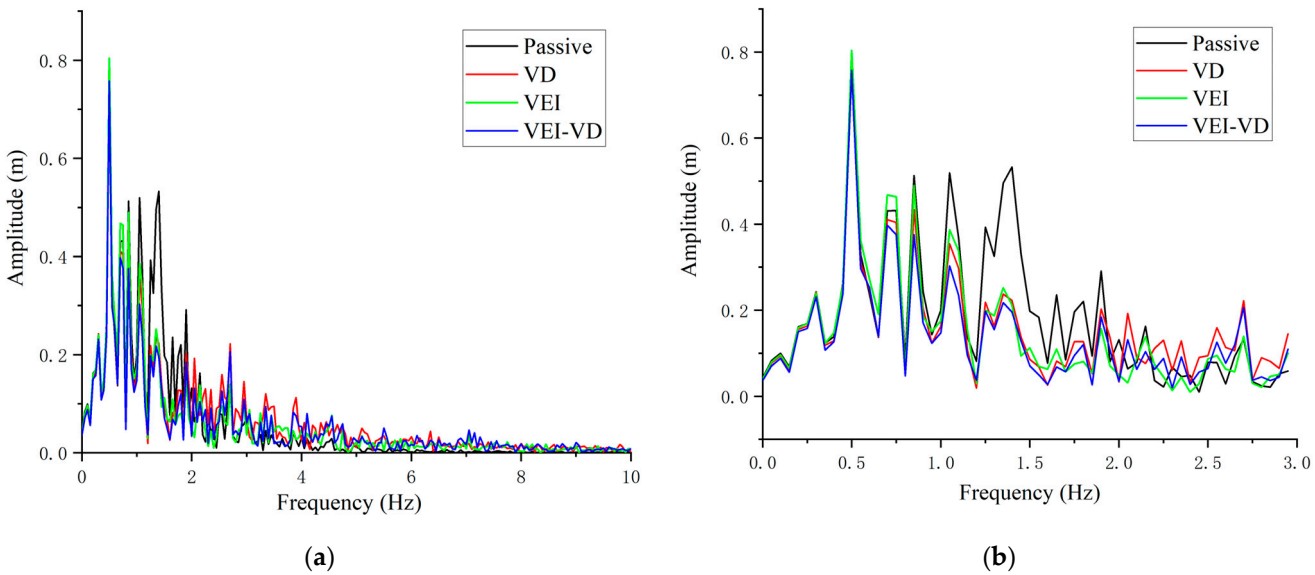

**Figure 18.** Acceleration in the frequency domain: (**a**) Entire experiment; (**b**) Zoom in.

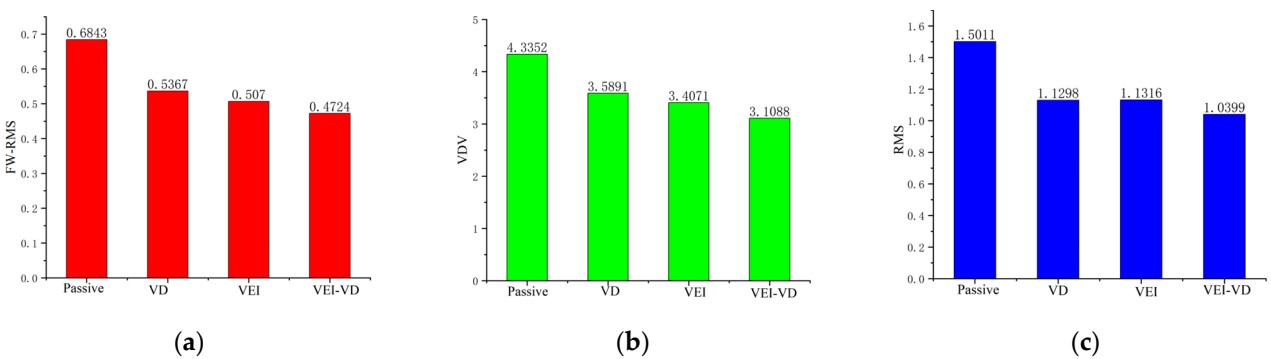

**Figure 19.** Evaluation parameters of seat acceleration: (**a**) FW-RMS; (**b**) VDV; (**c**) RMS.

**Table 3.** Vibration reduction compared to passive.

| Device | FW-RMS | VDV | RMS |
|--------|--------|-----|-----|
| Passive | — | — | – |
| VD | 21.57% | 17.21% | 23.28% |
| VEI | 25.91% | 21.41% | 24.62% |
| VEI–VD | 30.97% | 28.29% | 30.72% |

Figures 20 and 21 show the damping of the VEI–VD device and the external resistance of the circuit. It applied that the damping of the VEI–VD device can be controlled by controlling the resistance value of the external resistor, to change the equivalent inertance and equivalent damping of the VEI–VD device, then, the output force of the VEI–VD seat suspension can be controlled.

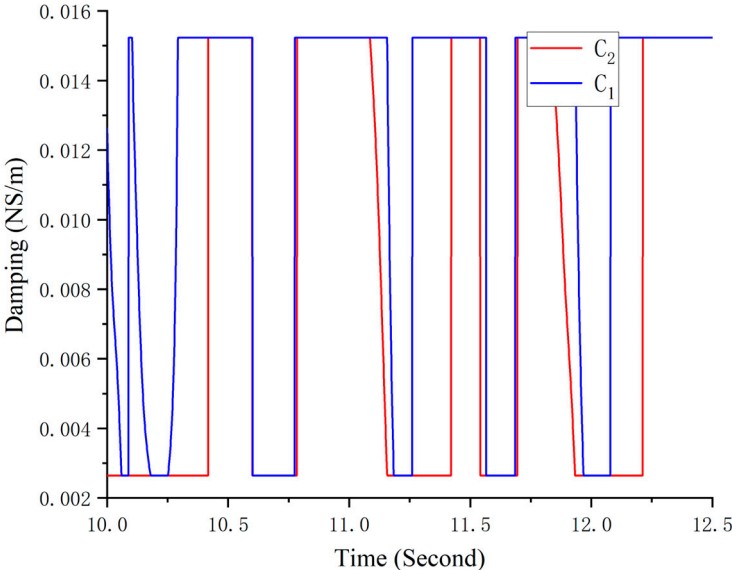

**Figure 20.** The damping of VEI–VD device.

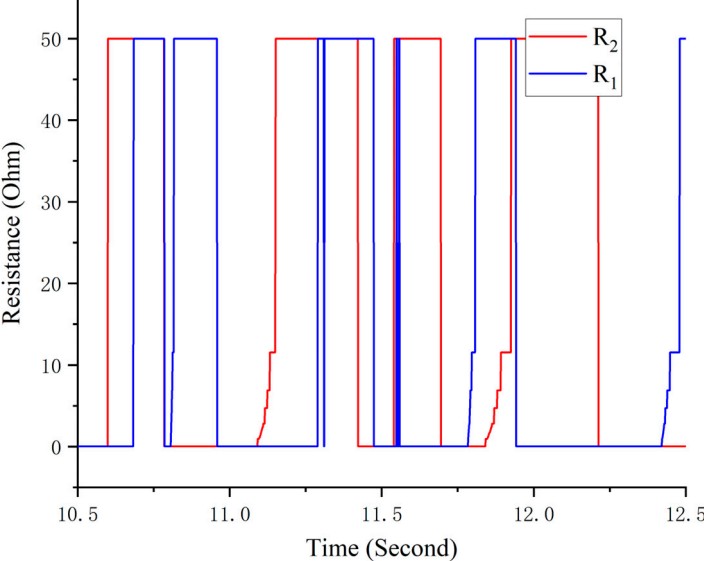

**Figure 21.** The resistance of VEI–VD device.

Figure 22 shows the power of semi-active seat suspension, the power is the product of the output force and output velocity of the suspension, it implies the VEI and VEI–VD

devices provided power to the outside. In VEI and VEI–VD seat suspensions, when the power of them is less than zero, it means that a part of the mechanical energy of seat vibration is stored in the flywheel. When the power is greater than zero, it means that the stored energy in the flywheel is converted into mechanical energy and suppresses the vibration of the suspension.

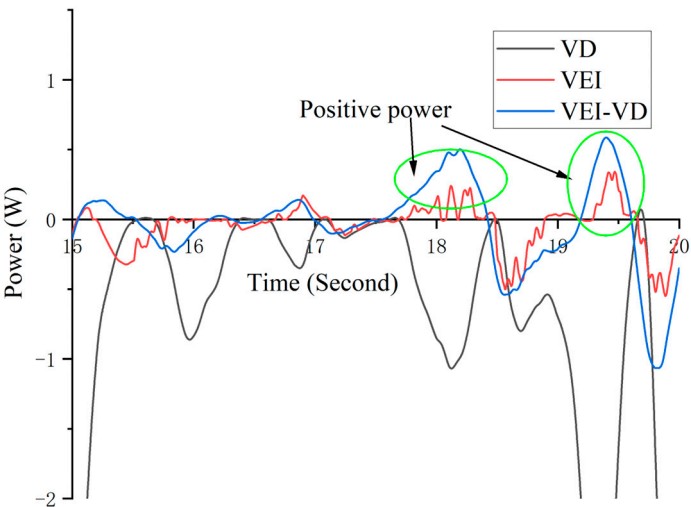

**Figure 22.** The power of semi-active seat suspension.

*5.4. Bump Test*

In addition, the vibration control performance of these seat suspensions under bump excitation is evaluated. When a vehicle passes a bump, the displacement of its sprung mass is used as the vibration input for these seat suspensions.

The Figure 23 shows that acceleration under bump excitation. The peak-to-peak acceleration difference of the VD, VEI and VEI–VD seat is reduced by about 21.33%, 25.43%, and 29.26%, respectively. Furthermore, the VEI–VD seat's acceleration using the semi-active control strategy at the second and third vibration peaks is better than that of the VD and VEI seat, indicating that the designed VEI–VD device has a better vibration control effect on the seat.

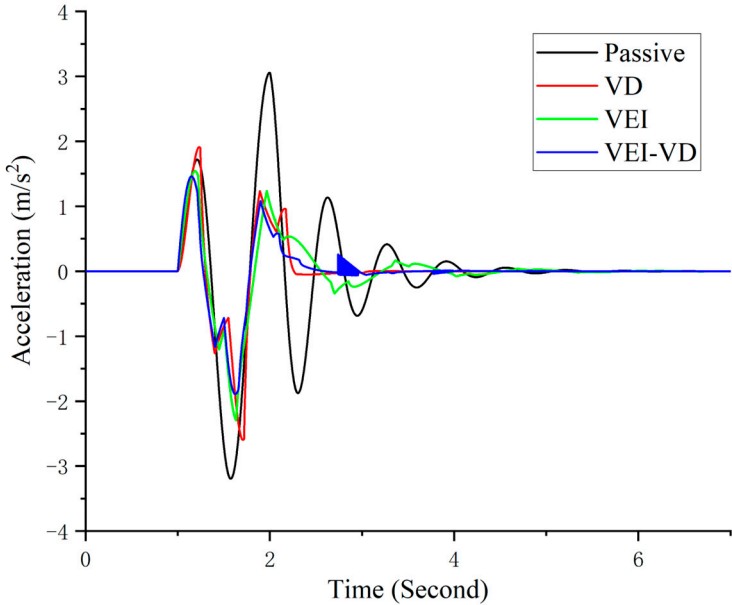

**Figure 23.** Acceleration under bump excitation.

## 6. Conclusions

This paper has proposed a VEI–VD device and a semi-active control strategy to verify its performance in vibration control. In the proposed device, the VEI part uses a flywheel as an inertance element, and both VEI and VD parts use EMDs to realize damping change. The dynamic model of the device has been proposed and verified with experiments. A semi-active control strategy that considers the energy storing and releasing of VEI–VD device has been designed to suppress the seat vibration. Under the random excitation test, the vibration reduction of VD, VEI, and VEI–VD seat compared with passive seat are 23.68%, 24.62%, and 30.72%, respectively. And in the bump test, the peak-to-peak acceleration is reduced. All tests indicate that the novel semi-active control strategy of seat suspension equipped with VEI–VD device has greater potential compared with other traditional semi-active control strategies. At this stage, we have developed and tested the prototype of the VEI–VD device, and its vibration control performance has been validated based on its model. However, the time-delay influence on its performance has yet to be studied, and a seat suspension with the proposed device needs to be built to verify its actual vibration control performance. In future work, we will optimize the system parameters of the VEI–VD device, improve the control method, and apply the device in practical application.

**Author Contributions:** Writing—original draft preparation, data curation: G.L. (Guangrui Luan); validation, methodology, G.L. (Guangrui Luan) and P.L.; writing—review and editing, funding acquisition, conceptualization, D.N.; project administration, resources, G.L. (Guijie Liu); supervision, H.D. All authors have read and agreed to the published version of the manuscript.

**Funding:** This research was funded by the National Natural Science Foundation of China (Grant No. 52088102) and the Nature Science Foundation of Shandong (Grant No. 2022HWYQ-067).

**Data Availability Statement:** No date need be reported.

**Conflicts of Interest:** The authors declare no conflict of interest.

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
