# Peer review of "Semi-Active Vibration Control of Seat Suspension Equipped with a Variable Equivalent Inertance-Variable Damping Device"

_machines, doi:10.3390/machines11020284_

Round 1

Reviewer 1 Report

The comments can be found in the attached file. 

Reviewer 2 Report

In the paper a prototype of variable equivalent inertance-variable damping device and its control system are presented. The article is interesting, the developed issue can be further developed and is inspiring for researchers interested in similar topics.  Despite the interesting subject matter, the presentation of the results requires significant improvement. I have some comments on the submitted manuscript that should improve the reading of this article.

1. Significant linguistic editing is required. The article contains a large number of editing, grammar and punctuation errors. There were also problems with the references sources used in the file.

2. Please also verify the unit symbols with the correct notation in the SI system, e.g. we use kg instead of Kg

3. In equation 11, saturation was used. Why and with what limits?

4. In table 1, the friction coefficient is dimensionless. Are you sure? Then the units in the force equations 11 and 14 don't match up.

5. Figures require quality improvement, especially Figures 12, 14 and 15 (no axis description)

6. The conclusions lack information about any limitations and problems at the stage of implementation and experiment. In the case of prototype projects, these are valuable information for the reader.

Round 2

Reviewer 2 Report

Dear Authors,

thank you for the effort you put into correcting the text. I believe the manuscript can be published.

As for the answer: if we assume that the value of 'sat' is dimensionless, then everything is correct now. However, this is quite a contentious issue. I suggest in lines 249 and 250 to remove the statement that it is "Coulomb friction", and leave only the phrases "friction coefficient" and "friction force". The formula rather indicates viscous friction, because we do not take into account the pressure reaction in it. However, the intention of the authors is clear to the reader.

Congratulations on your interesting research.
